# The Impact of Terrorist Attacks on Foreign Exchange Rate: Case Study of Turkish Lira versus Pound Sterling

**Mansoor Maitah [1,*], Jehar Mustofa [1] and Gok Ugur [2]**

[1] Department of Economics, Faculty of Economics and Management, Czech University of Life Sciences, Prague, Kamýcká 129, 165 00 Praha 6, Czech Republic; mustofa.seid@gmail.com

[2] Institute of Economics Studies, Faculty of Social Science, Charles University, Prague, Opletalova 26, 110 01 Praha 1, Czech Republic; uur.gok@gmail.com

[*] Correspondence: maitah@pef.czu.cz; Tel.: +420-234-382-139

**Abstract:** In this study, the impact of terrorist attacks on exchange rate is estimated. Particularly, the study focuses on terrorist attacks in Turkey and its implication on Turkish lira versus pound sterling exchange rate. In order to find the causal effect, the study employed Autoregressive distributive lag (ARDL) bound testing approach as an estimation technique. Accordingly, the analysis reveals that a terrorist attack has a negative impact on the exchange rate in both the short-run and long-run. However, the negative effect of terrorism tends to be small in both the short-run and long-run. More precisely, terrorist attacks depreciate the exchange rate between Turkish lira and pound sterling by approximately 0.024% in the next trading day. The long-term effect also shows that a terrorist attack depreciates the exchange rate on average by 0.0706%.

**Keywords:** terrorism; exchange rate; ARDL; error correction

## 1. Introduction

Terrorism is one of the rising global problems and the most widely discussed issues in recent years. However, according to the Global Terrorism Index Report (2015) [1] of The Institute for Economics and Peace (IEP), the majority of terrorist attacks were mainly concentrated in a small number of countries. During the last decade, Nigeria, Iraq, and Afghanistan have experienced the highest level of terrorism in the world. Nevertheless, countries like Turkey have been suffering from terrorism for an extended period. Moreover, Turkey is exposed to a high number of terrorist attacks every year. Historical statistics of the Global Terrorism Index Report (2014) [2] of The Institute for Economics and Peace (IEP) show Turkey experiencing the highest number of terrorist attacks within the OECD countries.

From an economic point of view, most of the empirical findings in the literature show that terrorist attacks induce a higher level of uncertainty and affect the economy negatively. There is a growing literature on the effect of terrorism on the economy in recent decades and many of these researchers find a negative impact of terrorism on the economy (Abadie and Gardeazabal, 2003 [3]; Enders and Sandler, 1996 [4]; Chen and Siems, 2004 [5]; and Suarez and Pshisva, 2006 [6]).

Considering the negative impacts of terrorist attacks on the economy, in this study we aim to investigate the effects of terrorist attacks on exchange rate, in particular Turkish Lira versus British sterling. The selection of British sterling as reference for the exchange rate is due to the size of British-based companies' investments in Turkey. According to the Turkish ministry of foreign affairs

office, there are more than 2800 British firms doing business in Turkey and their total business size is around 30 Billion Turkish Liras. Furthermore, trade between Turkey and the UK is around 45 Billion Turkish Liras and additionally the UK's direct investments in 2011 reached to 2 Billion Turkish Liras (Relations between Turkey and the United Kingdom [7]). With respect to increasing size of economic cooperation and relations between Turkey and the UK, in this research we seek to draw attention to possible negative effects of terrorist attacks because of increasing concern about terrorist incidents in recent years.

Hence, the study aims to provide an input to academia, policy makers, and business through identifying the relationship between terrorist attacks and the foreign exchange market, particularly Turkish lira versus pound sterling. Further, the paper contributes to policy makers and business by identifying both the short-term and long-term effect of terrorist attacks on the exchange rate. In fact, both policy makers and business could use the finding as an input for a policy intervention and trading strategy, respectively.

The rest of the paper is organized as follows. Section 2 provides the empirical and theoretical findings in the literature; Section 3 describes the data sources and empirical method; Section 4 presents and discusses our results; Section 5 concludes the paper.

## 2. Literature Review

Exchange rate determination and volatility is a vital subject in both macroeconomics and the currency market. However, forecasting the exchange rate remains a challenging task for both academicians and currency traders. Further, empirical findings from several exchange rate forecasting models fail to generate valid results, regardless the models are based on the advanced statistical framework and macroeconomic fundamentals. For instance, the seminal paper of (Meese and Rogoff, 1983 [8]), shows the structural exchange rate models applied in their article fail outdo the random-walk model in both short and long run. Similarly, Mussa, 1979 [9] finds that exchange rate changes are unexpected and follow a random-walk. In subsequent papers, (Meese and Rose, 1991 [10]) using non-linear and non-parametric models of exchange rate find negative results. Equally, Engel, 1994 [11] found that Markov-switching models outperform random walk model only in case of in-sample estimation, while the random walk outperforms the Markov-switching model when estimating out-sample fit.

According to (Sarno and Tylor, 2002 [12]), admitting the theory of exchange rate determination made conceivable models, empirical findings on exchange rate are yet to make a robust, statically sufficient and reliable model. Markedly, despite the exchange rate determination models occasionally accomplishing valid results in-sample, they often fail to out-perform the random walk out-sample forecasting. In a similar vein, (Bacchetta and van Wincoop, 2006 [13]) regard the weak performance of exchange rate determination theories and models as being the main weakness of international macroeconomics.

Therefore, the existing exchange rate determination models will not be able to capture the out-sample forecast for two main reasons. First, the stated exchange determination theories and models fail to capture the out-sample fit in previous research. Second, the exchange rate models are based on macroeconomic fundamentals. Hence, to capture the effect of terrorist attacks on exchange rate we have to go beyond the traditional exchange rate forecast theories and models.

In comparison to exchange rate studies, literature for terrorism related studies is not very abundant. Nonetheless, studies in this area have been expanding very rapidly due to the increasing concern about terrorism in the world. Terrorism incidents take different forms and terrorist motives can differ as well. In this respect, research papers in this area show diversifications in terms of focus area. For example, a study by Landes, 1978 [14] is one of the earliest studies in the evaluation of counter-terrorism policies in the literature. The author examines the installation of metal detectors at airports and the increase in the sentence on hijackings in the USA. Landes adapted (Ehrlich and Becker, 1972 [15]) the crime participation model into hijacking and used ordinary least squares (OLS)

to ascertain the effect of increasing security measures at the US airports. The author used quarterly data on hijacking events in the USA between the periods 1961–1976. His findings reveal that both an increase in sentence and security at airports has a deterrent effect on hijacking. Other similar studies by (Cauley and Im, 1988 [16], and Enders and Sandler, 1990 [17]) show that the installations of metal detectors at airports are likely to cause substitution or displacement effect. In other words, policies designed to reduce one type of terrorist attack are likely to increase the other types of terrorist attacks.

Apart from counter terrorism studies, researchers also interested to study the impacts of terrorism on the economy. Terrorist incidents impose economic costs to targeted country in a variety of ways. For example, a research study by Shahzad, et al., 2016 [18], evaluated the impact of terrorism on Foreign Direct Investment (FDI) on Pakistan. The authors find that terrorism has a deteriorating impact on FDI. Another study by Eldor and Melnick, 2004 [19] investigated the effects of terrorism on financial markets and the result of their study shows that intensified Palestinian terrorist attacks had a permanent negative effect on the stock market.

Despite the growing number of studies examining the effect of terrorist attacks on some key macroeconomic variables, the number of studies examining the implication of terrorist attack on the exchange rate is limited. Therefore, in this study, we examine the effect of terrorist attacks on exchange rate using Turkish lira and pound sterling. Through examining the causal effects of terrorism, we aim to contribute to the current literature and forward an input to policy makers, business and the society. In the next section, we have presented the estimation method and data employed to achieve the objective of the study.

## 3. Methodology

The autoregressive distributive lag (ARDL) bound testing approach has been used in this article. The ARDL approach is employed to examine the impact of terrorism attacks on exchange rates in Turkey. The extended ARDL approach by (Pesaran, et al., 2001 [20]) which is based on the estimation of Error Correction Model (ECM) is used in this study. Applying an ARDL model has several advantages, among others: First, it is possible to apply bound test because ARDL can be applied with small sample size study (Pesaran, et al., 2001 [20]). Second, it is possible to estimate both the short and long term relation while simultaneously solving autocorrelation and omitted variable bias. Third, according to (Harris and Sollis, 2003 [21]) the estimation gives unbiased result and the t-statistics are valid even in cases where some of the explanatory variables are endogenous. Fourth, it is possible to estimate the Cointegration relationship using ordinary least square (OLS) once we select correct lags in the ARDL model. Last but not least, variables used in the study do not need to be cointegrated in the same order.

Taking the advantages offered by ARDL bound testing approach, this study estimates the causality between terrorist attacks and exchange rate using the following equation:

$$\Delta lnEXC_t = \alpha_0 + \sum_{i=1}^{n} \alpha_{1i}\, \Delta lnEXC_{t-i} + \sum_{i=0}^{n} \alpha_{2i}\, \Delta lnTER_{t-i} + \alpha_3\, lnEXC_{t-1} + \alpha_4\, lnTER_{t-1} + \mu_t \quad (1)$$

where *lnEXC* is the log of the exchange rate between Turkish lira and pound sterling, *lnTER* represents the log of terrorist attacks in Turkey; while $\mu$ is disturbance term and $\Delta$ is first difference of the respective variables. The exchange rate data (EXC) are computed from bank of England data, while the data for TER are computed from Global terrorism database (https://www.start.umd.edu/gtd/).

In estimating the ARDL bound testing approach, first we have to estimate Equation (1) using ordinary least square (OLS) to examine the existence of long term relation through making joint *F*-tests. The joint *F*-test are made for the following null hypothesis, $H_0: \alpha_3 = \alpha_4 = 0$ and $H_0: \beta_3 = \beta_4 = 0$ and their respective alternative hypothesis and consequently two *F*-statistics are generated (Pesaran, et al., 2001 [20]). Accordingly, if the *F*-statistics is below the critical value, the null hypothesis of no long term Cointegration cannot be rejected. However, if the *F*-statistics are above the critical value we reject the null and accept the

alternative that is there is long term Cointegration. However, if the *F*-statistics fall within the critical value the finding will not be conclusive.

The second step in estimating the ARDL bound testing approach, after the Cointegration is determined, the conditional ARDL long run model for $EXC_t$ and $TER_t$ are estimated as:

$$lnEXC_t = \alpha_0 + \sum_{i=1}^{n} \alpha_1 \, lnEXC_{t-i} + \sum_{i=0}^{n} \alpha_2 \, \Delta lnTER_{t-i} + \varepsilon_t \tag{2}$$

All the coefficients represented here are similar to what already explained. The model also involves ARDL order selection using Schwarz Bayesian Criterion (SBC). The final step is estimating the error correction model (ECM) and finds the short run parameters. Hence, the ECM is estimated as:

$$\Delta lnEXC_t = \alpha_0 + \sum_{i=1}^{n} \alpha_1 \, \Delta lnEXC_{t-i} + \sum_{i=0}^{n} \alpha_2 \, \Delta lnTER_{t-i} + \varnothing ECM_{t-1} + \varepsilon_t \tag{3}$$

where $\alpha_1$ and $\alpha_2$ are short term coefficients of convergence of the model and $\varnothing$ represents the speed of adjustment. While the ECM coefficient is the error correction obtained from equilibrium relation of the estimation from Equation (1).

## 4. Estimation Result

The first step in applying the ARDL model is checking neither of the series are I(2). Therefore, to confirm both are not I(2) we used Augmented Dickey Fuller test and Schwarz Info Criterion with maximum lag. Accordingly, the ADF test to the levels of EXC and TER, the *p*-value shows 0.9061 and 0 respectively. Indicating while TER is stationary at level EXC fails to be stationary. However, after applying first difference on EXC series, the *p*-value becomes 0, indicating it is stationary. Therefore, the estimation in Table 1 shows that neither of the series are I(2).

**Table 1.** Unit root test results.

| Augmented Dickey-Fuller Test | | | |
|---|---|---|---|
| **Variables** | **Intercept** | **Intercept & Trend** | **Decision** |
| EXC | −9.241440 *** | −9.352369 *** | I(1) |
| TER | −10.57320 *** | −10.72092 *** | I(0) |

*Note:* Author's calculation (Eviews 9). *** Denotes statistical significance at 0.01 level. The variables EXC and TER represent sterling to lira exchange rate and number of terrorist incident respectively. The lag length is determined through Schwarz information Criterion. *Source:* the Global Terrorism Database (https://www.start.umd.edu/gtd/) and Bank of England Database (http://www.bankofengland.co.uk/Pages/home.aspx).

Then, since neither of our series are I(2) , we can apply Autoregressive Distributed Lag (ARDL) bounds testing approach to estimate the impact of terrorism on the exchange rate between sterling and Turkish lira. However, before we apply the Autoregressive Distributed Lag (ARDL) bounds testing approach, the appropriate lag length is selected using Schwarz Criteria (SC). Accordingly, Figure 1 reveals that ARDL (2,2) is the appropriate model.

The bound test in Table 2 shows that the computed *F*-statistics (51.101478) is greater than the upper bound of 7.84 at one percent level. Hence, we can reject the null hypothesis that no long-run relationships exist, accepting the alternative there exists a long-run co-integration relation between terrorism incidents and exchange rate in Turkey.

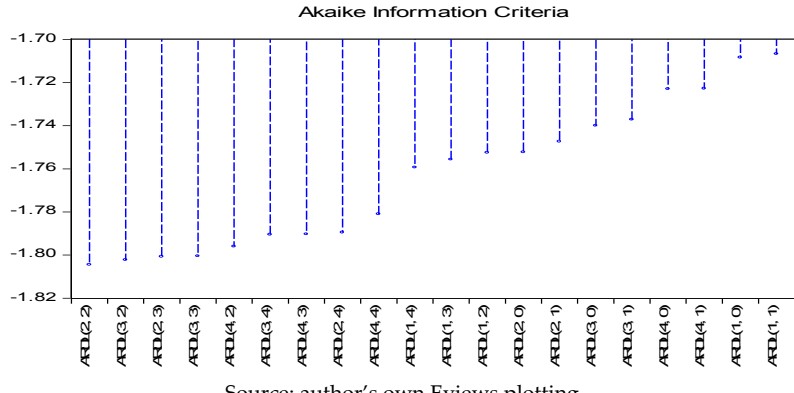

Source: author's own Eviews plotting.

**Figure 1.** Schwarz criteria.

**Table 2.** Autoregressive distributive lag (ARDL) bounds test.

| Null Hypothesis: No Long-Run Relationships Exist | | |
|:---:|:---:|:---:|
| **Test Statistic** | **Value** | ***k*** |
| *F*-statistic | 51.01478 | 1 |
| **Critical Value Bounds** | | |
| **Significance** | **Lower Bound** | **Upper Bound** |
| 10% | 4.04 | 4.78 |
| 5% | 4.94 | 5.73 |
| 2.5% | 5.77 | 6.68 |
| 1% | 6.84 | 7.84 |

*Note:* Author's calculation (Eviews 9). The lower and upper bound values are obtained from Pesaran, et al., 2001 [20], p. 300. The *F*-statistic is the computed value. *Source:* the Global Terrorism Database and Bank of England Database.

After confirming the existence of Cointegration relation between the covariates, the long-run coefficients for the selected ARDL (2,2) are presented in the following table. The result in Table 3 shows that the estimated coefficient of terrorism is significant and negatively affects the exchange rate. Accordingly, the result shows that one terrorist attack negatively affects the exchange rate by 0.024 percentage points, all things being the same. The empirical finding we got confirms that terrorist attacks have a long lasting effect of Turkish lira exchange rate with British sterling.

**Table 3.** Long-run coefficients using ARDL (2,2).

| Dependent Variable: D(EXC) | | | | |
|:---:|:---:|:---:|:---:|:---:|
| **Variable** | **Coefficient** | **Std. Error** | ***t*-Statistic** | **Prob.** |
| D(TER) | −0.002117 | 0.000663 | −3.192802 | 0.0018 |
| C | 0.010292 | 0.007391 | 1.392620 | 0.1666 |

*Note:* Author's calculation (Eviews 9). *Source:* the Global Terrorism Database and Bank of England Database.

The short-run dynamics coefficients from the estimation of the ARDL ECM are shown in Table 4. Similarly, to our previous lag selections the error correction lag for the ARDL (2,2) are selected through Schwarz criteria (SC). As can be seen in table, the estimated error correction coefficient is significant at the one percent level and has a negative sign. This indicates that the adjustment is not only slow but tends towards divergence from the initial level. Specifically, in the current period the exchange rate tends to further diverge from the log-run equilibrium level. Since our error correction coefficient is significant at one percent level, the finding confirms there is a short-run negative impact of terrorism on the exchange rate between Turkish lira and British pound.

**Table 4.** Error correction estimation for estimated ARDL (2,2).

| Dependent Variable: D(EXC) | | | | |
|---|---|---|---|---|
| **Variable** | **Coefficient** | **Std. Error** | *t*-**Statistic** | **Prob.** |
| D(EXC(−1), 2) | 0.243723 | 0.089068 | 2.736373 | 0.0073 |
| D(TER, 2) | −0.000715 | 0.000308 | −2.322655 | 0.0221 |
| D(TER(−1), 2) | 0.000899 | 0.000311 | 2.892511 | 0.0046 |
| ECM(−1) | −1.207987 | 0.122467 | −9.863758 | 0.0000 |
| **ECM = D(EXC) − (−0.0021 × D(TER) + 0.0103)** | | | | |
| *R*-squared | 0.144638 | Akaike info criterion (AIC) | −1.812668 | |
| *F*-statistic | 3.652465 | Schwarz criterion (SC) | −1.668658 | |
| Prob(*F*-statistic) | 0.004309 | Hannan-Quinn criter. (HQ) | −1.754223 | |
| Durbin-Watson stat | 1.876302 | | | |

*Note:* Author's calculation (Eviews 9). *Source:* the Global Terrorism Database and Bank of England Database.

Last, in Figure 2 we plot the cumulative sum of recursive residuals (CUSUM) to check the stability of our finding from the estimation of both long and short-run parameters from the ARDL (2,2) model with error correction. Accordingly, the CUSUM plot shows that the statistics remains with the bound of the five percent significance level. Therefore, we cannot reject the null hypothesis that all the coefficients are stable.

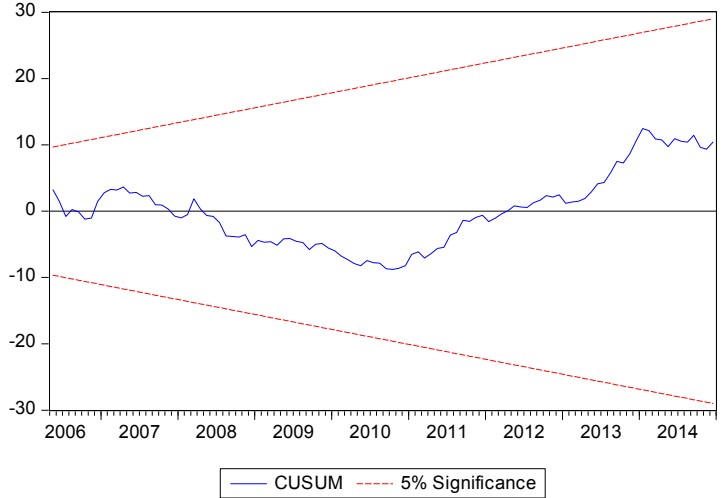

*Source:* the Global Terrorism Database and Bank of England Database.

**Figure 2.** Plot of cumulative sum of squares of recursive residuals—CUSUM of Squares Test for Turkey.

## 5. Conclusions

The main objective of the study is to examine the impact of terrorist attacks on exchange rate in the case of Turkey. In addition, the paper aims to contribute to academia, policy making and business. More specifically, the paper aspires to contribute to the existing research through widening the dimension of the research on the effect of a terrorist attack. In fact, to the author's knowledge this study is unique in identifying the effect of terrorist attack on exchange rate between Turkish lira and pound sterling. Further, the paper aims to provide an input to both policy makers and business in formulating policy intervention and trading strategies respectively.

Therefore, in order to achieve the objective, the study employs the autoregressive distributive lag (ARDL) bound testing approach as an estimation technique. Accordingly, the analysis reveals that a terrorist incident has a negative impact on the exchange rate in both the short-run and long-run. However, the negative effect of terrorism tends to be small in both the short-run and long-run. More precisely, the short-run effect seems much lower than the long-run effect of terrorism on exchange

rate. For instance, if everything remains constant, a one-person causality terrorist attack results in depreciation of Turkish lira versus pound sterling by 0.024% the next trading day. Further, a one-person causality terrorist attack has a depreciation effect in the next month by 0.0706%.

Although the depreciation seems small per one causality person when considering the historical number of the terrorist attack causalities in Turkey, the impact becomes large. For instance, if everything remains constant, a terrorist attack with casualty of 100 people could result in depreciation of Turkish lira by approximately 2.4% against pound sterling. The significance of this depreciation could be understood when considering the implication on foreign trade, servicing foreign debt, stock and the financial market. More specifically, a terrorist attack with 100 casualties could make domestically produced commodities cheaper in the international market and make it expensive to import foreign goods, resulting in inflation. Similarly, the same terrorist attack could make servicing pound sterling loans expensive.

In a nutshell, the present study provides a valuable policy and business strategy input for policy makers and business, respectively. For instance, policy makers could use the result to eliminate the inflation pressure resulting from the depreciation of Turkish lira and business to either maximize/minimize their profit/loss.

Last but not least, although the average effect of terrorist attacks on Turkish lira is negative, in some incidents we see the exchange rate either remaining constant or appreciating (Appendix A Figure). This could be due to the location of the city in which the terrorist attack happened and the media coverage given. For instance, if the attack is in tourist destination cities and subject to high media coverage, we expect the exchange rate to depreciate significantly. However, if the terrorist attack is not in a tourist destination city, we expect the media coverage and so the effect on exchange rate to be low. Therefore, capturing the effect of location and media coverage requires the use of panel data and panel estimation technique.

**Acknowledgments:** Mansoor Maitah and Jehar Mustofa mention that this article is performed within the internal grant No. 20151031, provided by the Internal Grant Agency of the Faculty of Economics and Management, Czech University of Life Sciences Prague.

**Author Contributions:** Jehar Mustofa and Ugur Gok contributed to data collection and management, and interpreted the results; Mansoor Maitah contributed to the analysis of the estimation results; Jehar, M. and Gok, U. provided analytical materials and methodological tools; Maitah, M., Jehar, M. and Gok, U. wrote the manuscript. All authors read and approved the final manuscript.

**Conflicts of Interest:** The authors declare no conflict of interest.

**Appendix A**

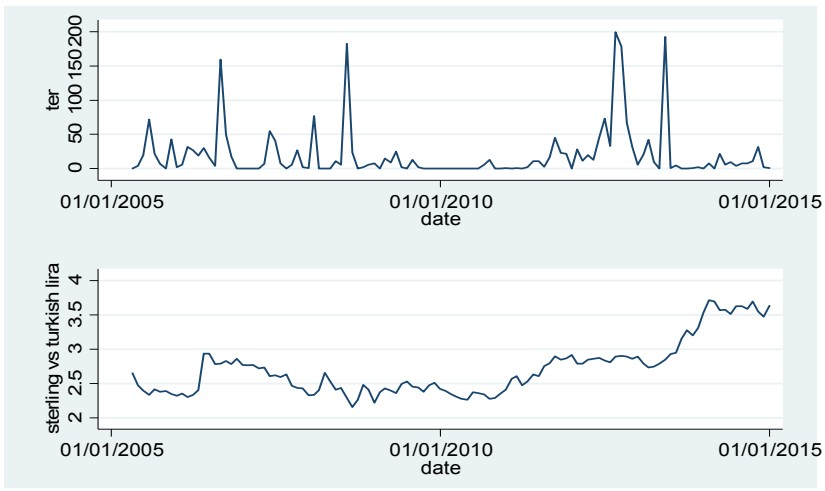

*Note:* Author's own plotting stata. *Source:* Bank of England Database.

**Figure A1.** Terrorist attack and Turkish lira vs. pound sterling exchange rate.

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
