# Peer review of "The Impact of Terrorist Attacks on Foreign Exchange Rate: Case Study of Turkish Lira versus Pound Sterling"

_economies, doi:10.3390/economies5010005_

Round 1

Reviewer 1 Report

 This is an interesting and topical issue addressed in this paper. It uses the ARDL bounds test for cointegration between the exchange rate and terrorism. The results suggest that terrorism causes a depreciation of the exchange rate. I have the following comments:

- In the literature review you could remove the results form the studies using the ARDL model where terrorism isn't included in the paper, although retain the discussion of this technique in the literature.

- You could include some reference to the exchange rate literature, for instance refer to a paper or papers on the effects of risk/ risk premium on the exchange rate.

- I would include the plots of the data in the text rather than the appendices, then discuss what has caused the changes to these variables.

- When doing the ARDL test for cointegration, could the authors clarify whether the variables were entered in levels or differences, as from the results, for the long-run coefficients it says D(EXC) and D(TER), does the D refer to differenced? The test statistic is high and could be due to having differenced the data before testing for cointegration.

- In the methodology there is a discussion of model 2. where the exchange rate causes terrorism, but no results on this. This section could be removed unless there is a reason for causality to go from the exchange rate to terrorism.

- Where the study describes the degree of effects between terrorism and the exchange rate, should the values be in percentage form, as the variables are logged?

Minor Point:

 You don't need to put 'Authors own computation' under the tables of results.

Author Response

Dear sir/madam, 

First i would like to thank you for your valuable review. Next, the comments you gave us are very important made adjustments accordingly.

the literature is adjusted to include exchange rate empirical works and excluded ARDL empirical works which has nothing to do with terrorism.

as you have already noted there was no need of the second model and therefore we took it out.

the results are interpreted in percentage.

with regards 

Reviewer 2 Report

The paper studies an interesting topic. However, the paper should make a significant reversion and more convincing explanations before ready for publication. I would like to make a few comments that will hopefully improve the exposition and discussion of the results.

The reason that the authors study the exchange rate of Turkish lira versus Pound sterling is not convicing. The authors should provide supportive data about the importance of number of tourists from UK on the bilateral exchange rate. How important does the number of tourists from UK affect Turckish economy.

Although the topic and content analyze the impact of terrorist attacks on Turkish lira versus Pound sterling exchange rate, but the model use lnEXC (the log of the exchange rate between Turkish lira and US dollar) to estimate.

The data used is not presented clearly. What is the sample frequency and the sample coverage?

It will be better to show the percentage effects on the exchange rate instead of the level effect.

The literature review is too long. The authors should drop those that are not directly related to the study.

Author Response

Dear Sir/madam,

First i would like to thank you for your valuable review and mention the adjustment we made on the article.

The literature is refined to include previous exchange determination literature and exclude ARDL empirical works which are not related with terrorism.

The data coverage and source are included in the article.

The result are presented in percentage points.

In the Methodology part model 2 is excluded because it was not relevant to the study.

with regards

Round 2

Reviewer 2 Report

I found this revision much clearer and significantly improved relative to the original submission, and I sincerely appreciate the effort the author put in addressing my comments. My recommendation to the editor is for an acceptance decision.